# A human monoclonal antibody binds within the poliovirus receptor-binding site to neutralize all three serotypes

Andrew J. Charnesky[1,2], Julia E. Faust[3], Hyunwook Lee [2,3], Rama Devudu Puligedda[4], Daniel J. Goetschius [1,2], Nadia M. DiNunno[1,2], Vaskar Thapa[3], Carol M. Bator[2], Sung Hyun (Joseph) Cho[2], Rahnuma Wahid[5], Kutub Mahmood[5], Scott Dessain[4], Konstantin M. Chumakov[6], Amy Rosenfeld[6] & Susan L. Hafenstein[1,2,3,7] ✉

Global eradication of poliovirus remains elusive, and it is critical to develop next generation vaccines and antivirals. In support of this goal, we map the epitope of human monoclonal antibody 9H2 which is able to neutralize the three serotypes of poliovirus. Using cryo-EM we solve the near-atomic structures of 9H2 fragments (Fab) bound to capsids of poliovirus serotypes 1, 2, and 3. The Fab-virus complexes show that Fab interacts with the same binding mode for each serotype and at the same angle of interaction relative to the capsid surface. For each of the Fab-virus complexes, we find that the binding site overlaps with the poliovirus receptor (PVR) binding site and maps across and into a depression in the capsid called the canyon. No conformational changes to the capsid are induced by Fab binding for any complex. Competition binding experiments between 9H2 and PVR reveal that 9H2 impedes receptor binding. Thus, 9H2 outcompetes the receptor to neutralize poliovirus. The ability to neutralize all three serotypes, coupled with the critical importance of the conserved receptor binding site make 9H2 an attractive antiviral candidate for future development.

The scientific study of poliovirus, the causative agent of poliomyelitis, has a long history going back over a century[1]. A poliomyelitis-like illness was depicted in ancient Egypt[2] indicating that the disease has been with humanity for millennia. The route of virus infection is fecal-oral transmission, with most cases being asymptomatic. Gastrointestinal symptoms occur in 4–8% of patients[3]. Cases of acute flaccid paralysis (AFP), the most severe outcome of a poliovirus infection (<1% cases)[3], are used as a metric for quantifying cases of poliomyelitis[4]. Three serotypes are recognized and designated as types 1, 2, and 3 that are separated by differing sequences and conformation

of epitopes[5]. Immunity to one serotype does not confer immunity to the other serotypes and protection against all three is required to prevent disease. Global eradication efforts are ongoing and have thus far successfully removed wild-type serotypes 2 and 3 from circulation; however, unfortunately, wild virus type 1 remains endemic to Afghanistan and Pakistan[6]. A significant obstacle to eradication is vaccine hesitancy and the reliance on vaccination campaigns. Further exacerbating efforts is a reliance on using the attenuated Sabin OPV that possesses the risk of reversion, including reacquiring neurovirulence and the development of vaccine-derived paralytic

[1]Molecular, Cellular, and Integrative Biosciences Program, The Pennsylvania State University, University Park, PA, USA. [2]Huck Institutes of the Life Sciences, The Pennsylvania State University, University Park, PA, USA. [3]Department of Biochemistry, The Pennsylvania State University, University Park, PA, USA. [4]Lankenau Institute for Medical Research, Lankenau Medical Center, 100 East Lancaster Avenue, Wynnewood, PA 19096, USA. [5]Center for Vaccine Innovation and Access, PATH, Seattle, WA 98121, USA. [6]Office of Vaccines Research and Review, Division of Viral Products, Laboratory of Method Development, FDA, Silver Spring, MD, USA. [7]Department of Medicine, The Pennsylvania State University College of Medicine, Hershey, PA, USA. ✉e-mail: shafenstein@psu.edu

poliomyelitis (VDPV)[7]. Cases of circulating VDPV (cVDPV) are currently present from all three serotypes[8] and can be spread in under-immunized populations as seen recently in London, the Netherlands, and New York State[9–11]. Thus, next-generation vaccines, antivirals, and biologics are critically important for the final eradication of wild and vaccine-derived poliovirus.

Poliovirus is a member of the genus *Enterovirus* within the family *Picornaviridae*. The viral capsid is about 30 nm in diameter and is comprised of four structural capsid proteins (VP 1–4) with $T = 1$ (pseudo $T = 3$) icosahedral symmetry[12]. Structural features of the capsid include the raised plateau, or mesa, at the fivefold axis of symmetry. Encircling each fivefold vertex is a depression called the canyon, which is the binding site for the poliovirus receptor (PVR)[13–15]. At the bottom of the canyon is an opening into a hydrophobic pocket in which resides a lipid molecule called the pocket factor (PF), which stabilizes the capsid[5,16]. The propeller is centered on the threefold axis of symmetry and is made up of slightly raised density that peaks at the propeller tip, or puff, on the edge of the canyon opposite the mesa.

The first high-resolution 3D structure of poliovirus allowed visualization of conformational antigenic epitopes previously recognized using escape mutations[12,17–23]. Neutralizing antigenic (N-Ag) site I through III were identified and mapped to the capsid. N-Ag I is at the north canyon rim, N-Ag II spans the region between the threefold axis of symmetry and the south canyon rim, and N-Ag III is centered on the puff. More recently, a chimpanzee-human chimeric antibody and nanobody footprints were mapped to the virus by solving the cryo-EM maps of the complexes and fitting or building into the structures to identify the contacts[24,25].

CD155 was identified as the PVR for all three serotypes at the end of the 1980s[26]. Modest 21–22 Å resolution 3-D structures of the PVR-capsid complex were used to interpret receptor binding by fitting homology models[13,14] and a recent 4 Å structure of the PVR-capsid complex clearly identified the footprint. This high-resolution structure of the complex displayed conformational changes of VP1 near the hydrophobic pocket and revealed the absence of pocket factor[15]. Interaction with PVR is the first critical step in virus entry used by all three serotypes of poliovirus. The current model of virus entry suggests that the binding of receptors at the canyon causes the release of the stabilizing lipid PF, allowing significant conformational changes. These changes result in the formation of a necessary entry intermediate called the altered particle, leading subsequently to the release of the genome. Since the receptor-binding site is an essential poliovirus active site that is conserved across serotypes, it is an attractive target for antivirals to cross-neutralize poliovirus serotypes[27–32].

Previously human mAb 9H2 was characterized and shown to neutralize poliovirus serotypes 1, 2, and 3[33]. Here, we solved the high-resolution structures of 9H2 fragments of antibody binding (Fab) complexed with live and inactivated poliovirus serotypes 1, 2, and 3 capsids. The resulting six maps had sufficient resolution (2.5–3.2 Å) to build the Fab variable domain and virus capsid without ambiguity, illustrate the conformational epitope, identify the Fab-virus contacts, and map the footprints. These studies identify a fourth neutralizing site within the canyon. The mode of binding was nearly identical among all the complexes with the Fab binding across and into the canyon. Despite pocket factor not being present in all versions of the 9H2 Fab-poliovirus complex, there were no conformational changes observed in the virus capsid. Several of the residues from the 9H2 footprint overlap with the receptor-binding domain. Thus, the most likely mechanism of virus neutralization is sterically blocking the PVR.

## Results

### 9H2-virus complexes were produced for cryo-EM

To make complexes, 9H2 antibody fragments (Fab) were generated and purified from the mAb and incubated in excess with virus capsids of wild-type PV1 and 2 (WTPV1, WTPV2), Sabin PV3 (SPV3), and Sabin inactivated PV1, 2, and 3 (SIPV1, SIPV2, SIPV3). Each mixture was incubated at room temperature for 30 min, screened by negative stain TEM, and vitrified on continuous carbon-coated Quantifoil grids for cryogenic electron microscopy (cryo-EM). Data were collected in-house at the Penn State University, Huck Institute's cryo-EM facility (Supplementary Table 1). In the micrographs for all samples, Fab density was visible decorating the capsid and most complex particles appeared intact with electron-dense central densities corresponding to the genome (Supplementary Fig. 1). In some samples unbound Fab was noticeable in the background.

### The reconstruction processing pipeline was uniform for each complex

After particle picking, each reconstruction proceeded to 2D classification with the best classes used for a subsequent homogeneous icosahedral refinement. The 3D refinement showed that the Fab bound across the canyon, with enough room between binding sites to accommodate all symmetry-related copies of the Fab without steric collision (Fig. 1 and Supplementary Fig. 2). Fab density was strongest adjacent to the capsid surface and density diminished further away from the binding interface. Local resolution mapping mirrored this trend, with the best resolution in the virus capsid and at the binding interface. The poorest local resolution was found at the apex of the bound antibody fragment (Fig. 2 and Supplementary Fig. 3). This result was likely due to the flexibility between the Fab variable and constant domains. For several of the complex maps, there was some weak Fab density in the interface that hindered efforts to build Fab de novo (Fig. 1 and Supplementary Fig. 2). To improve density quality for the bound Fab, Icosahedral Subparticle Extraction and Correlative Classification (ISECC)[34] was used for all the maps (Supplementary Fig. 4). Each sub-volume was designated to include the Fab and the capsid epitope. All symmetry-related positions on each capsid were extracted and classified, with selected classes post-processed with DeepEMhancer[35]. In all six cases, the subparticle approach improved the density. The resulting Fab variable domain densities at the binding interface were clearly resolved at final resolutions ranging from 2.5 to 3.2 Å (Supplementary Fig. 4) allowing for de novo Fab building.

### Refinement of virus and Fab was done using the high-resolution subparticles

For the virus, the build into the subparticle was initiated with the corresponding structure (PDB ID 1HXS [https://www.rcsb.org/structure/1HXS][36], 1EAH [https://www.rcsb.org/structure/1EAH][37], and 1PVC [https://www.rcsb.org/structure/1PVC][5]) for the WTPV1, WTPV2, SPV3 and SIPV3, respectively. Where the capsid structures were unavailable (Sabin PV1 and PV2) the wild-type structure (PDB ID 1HXS and 1EAH) was used with modification to include all point mutations, insertions, and deletions. A Fab model was calculated by submitting the 9H2 amino acid sequence to the SabPred AbodyBuilder[38]. This 9H2 model initiated the build that was guided by the presence of disulfide bond density visible in the WTPV2, SPV3, SIPV1, and SIPV2 complex maps. Fab binding relative to the capsid appeared comparable in each complex where Fab can be seen binding into and spanning the canyon (Fig. 1, Supplementary Fig. 2, Supplementary Movie 1).

### There were no gross conformational changes to the virus

To assess possible conformational changes initiated by Fab binding to live virus, the final refined virus model was superimposed with the corresponding crystal structure used to initiate the build. The α-carbon root mean square deviation (Cα-RMSD) was 0.7, 0.4, and 0.4 Å for live virus serotypes 1, 2, and 3, respectively. Strong pocket factor density, equal in magnitude to capsid density was observed in both inactivated and live types 2 and 3 virus-Fab complexes; however, there was no PF density resolved at any contour level in either the inactivated or live type 1 virus-Fab complexes. The N-terminal VP1 residues of all

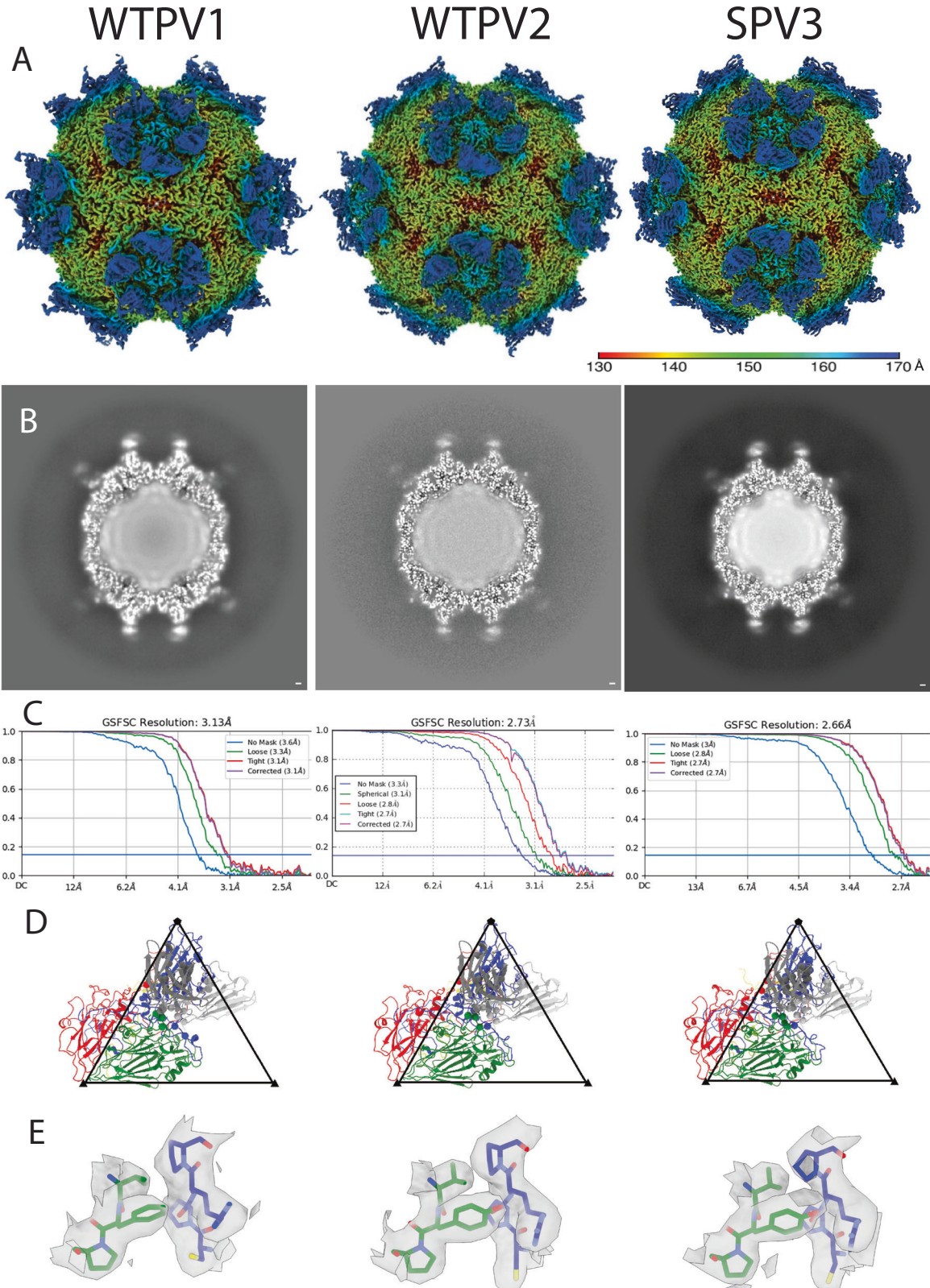

**Fig. 1 | 9H2 Fab binds virus with the same mode of binding in all three serotypes. A** Surface rendered DeepEMhancer sharpened maps are colored according to radius (color key) and show density corresponding to the 9H2 Fab. **B** The central section shows the Fab has a similar magnitude of density as the capsid. **C** FSC curves indicate resolution ranging from 2.7–3.1 Å at the gold standard 0.143 cutoff. **D** Refined models of single virus protomer (VP1–4; blue, green, red, yellow) and bound 9H2 variable domain (heavy and light chain; dark and light gray). Live virus 1, 2, and 3 data were taken at magnification x59,000, x59,000, and x120,000 resulting in pixel sizes of 1.1, 1.1, and 1.2, respectively. **E** For each complex, a representative area illustrating the quality of the model built into the map is shown. The map area highlighted includes residues VP1 270–273 and VP2 192–194. Coloration is VP1 (blue) and VP2 (green) with additional coloring by heteroatom.

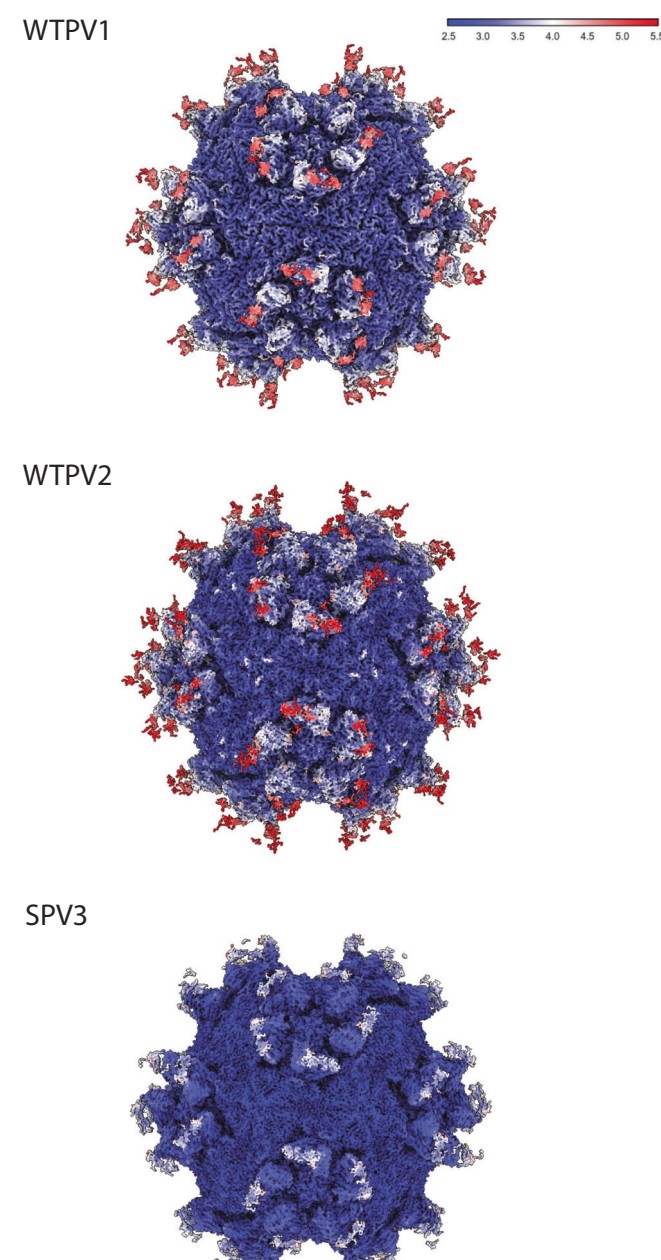

WTPV1

WTPV2

SPV3

**Fig. 2 | Local resolution maps.** Local resolution values were measured in Angstroms and displayed with a surface rendered map for each live virus-Fab complex. The best resolution is present in the capsid shell while the poorest is found in the hinge region of the Fab fragments.

complexes aligned with no changes. One local difference was seen in WTPV1 at a VP1 loop comprised of residues 232–238. In the crystal structure 1HXS, this loop forms a short helix, but the curvature of the loop for the Fab-WT PV1 complex follows closely with the crystal structures of the other two serotypes and the Fab-virus complexes made from PV2 and SPV3[36] (Supplementary Fig. 5). Another crystal structure of PV1 (PDB ID 1ASJ) [https://www.rcsb.org/structure/1ASJ][39] was superimposed and showed the 232–238 loop followed the consensus, suggesting that the loop difference was due to crystallization conditions or packing for 1HXS[36].

Another noticeable difference in local conformation was seen in PV2 and SPV3 VP1 loop 96–103 at the fivefold, which is located within the Fab binding interface. For PV2, both for the Fab-PV2 complex map and the X-ray crystal structure the loop was disordered. In inactivated

PV2 complexed with Fab, the loop has continuous density and residues Arg100 and Ala101 were assessed as contacts. In live SPV3 complexed with Fab, there is discontinuous density that cannot be built between VP1 residues 98 and 101; however, the entire loop can be visualized in the inactivated PV3 complexed with Fab, where Arg100 was identified as a contact. For the crystal structure, the VP1 96–103 loops are built, but do not align closely with the inactivated virus structures (Fig. 3). This finding suggests that in live SPV3, the binding of Fab may cause local disordering at this VP1 96–103 fivefold loop.

### Contact identification

The Fab binding relative to the surface of the virus appeared nearly identical for each complex. The refined Fab structures superimposed with Cα-RMSD of 0.3–0.5 Å compared to WTPV1 bound Fab. For each Fab-virus complex the corresponding refined Fab structure was used to identify the contact residues in the binding interface (Fig. 4) and the footprint was mapped (Fig. 5). To compare footprints, PV2 and 3 sequences were aligned to PV1 and the amino acid numbering used throughout is according to PV1 sequence-equivalent residues.

In all complexes, 9H2 Fab makes contact with VP 1, 2, and 3 (Table 1 and Supplementary Table 2). Total 9H2 contacts on the virus surface range from 23–27 residues (Supplementary Fig. 6). The buried surface area ranges from 1200–1400 Å$^2$. There are eleven 9H2 Fab contacts that are common to all complexes, which were termed universal contacts. Only seven of these universal contacts (VP1 V87, I89, F105, W108, D114, L228, and VP3 A235) on the capsid surface are sequence conserved among the virus used to make all six complexes.

Among all 9H2 contacts identified in the complexes, 14 are also in the poliovirus receptor-binding site. Five of these residues are universal contacts: VP1 105, 106, 107, 168, and 228. Of these five, only VP1 F105, and L228 are conserved in sequence identity. Among the live virus 9H2 footprints, there is one unique contact for serotype 1, VP1 L234, that overlaps with the receptor-binding site. All variants of the 9H2 footprint make contact on both sides of the canyon as well as with some residues of the canyon floor (Fig. 5 and Supplementary Movie 1). Inactivated virus footprints are not significantly different than the corresponding live serotype although we cannot assess the contact details completely within the disordered VP1 loop 96–103.

### Comparison of live versus inactivated virus

A comparison of the refined structures of live and inactivated virus structures showed no significant differences. The 96–103 and the 232–238 loops superimpose completely in each comparison for resolved residues. For types 1 and 2, the live wild-type virus superimposed with inactivated Sabin with a Cα-RMSD of 0.3 and 0.3, respectively. Live Sabin 3 superimposed with inactivated Sabin 3 with a Cα-RMSD of 0.4. There were differences identified for the Fab contacts; however, most are flanking conserved contacts (Supplementary Fig. 6).

### Fab binding

The refined virus structures were used to identify the Fab residues in the interface. Most virus contacts mapped to the Fab heavy chain, varying from 11–13 residues among the complexes. Fab light chain residues were less involved with only 6–10 residue interactions identified across all CDRs (Table 2 and Supplementary Table 3). Heavy chain CDR loop 3 defined by Paratome[40,41] made the most contacts in every complex and reached down into the canyon making contact with the north canyon rim and canyon floor. Only 1-2 contacts were identified among the six complexes on the other heavy chain CDRs and light chain contacts were evenly distributed among light chain CDRs. Fab contact residues were more conserved than viral contacts.

**9H2 neutralization, affinity, and competition with receptor.** The efficacy of 9H2 Fab neutralization was described in Puligedda et al. 2017[33] where it was reported that the reciprocal of the dilutions that

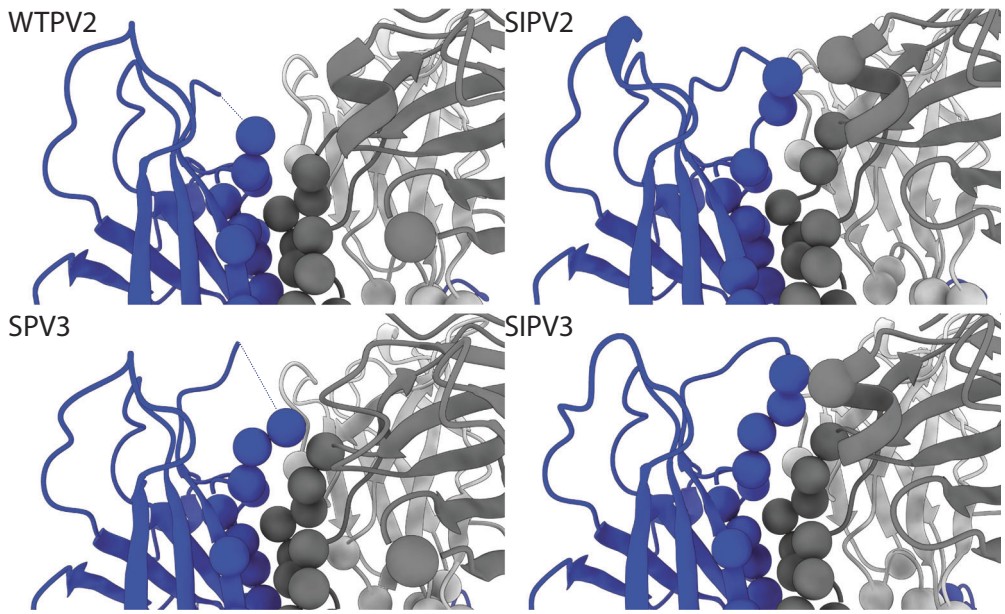

**Fig. 3 | A Mesa Loop is Flexible in WTPV2 and SPV3.** VP1 residues 96 to 103 are disordered in the WTPV2 and SPV3 complexes in contrast to the inactivated complexes of the same serotype. Residues identified as contacts are displayed as spheres matching their chain color (heavy and light chain; dark and light gray; VP1; blue).

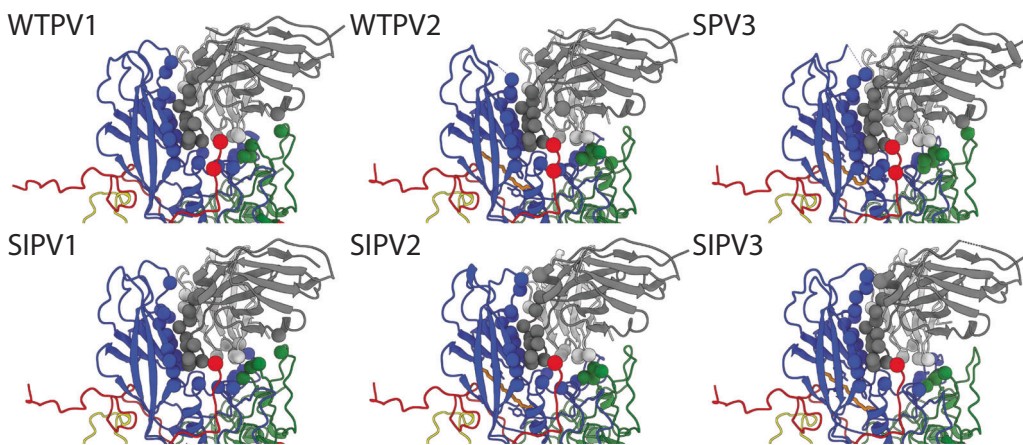

**Fig. 4 | 9H2 model contacts.** View is looking down tangent to the canyon at individual 9H2-poliovirus complex protomer models. Residues identified as contacts are displayed as spheres matching their chain color (heavy and light chain; dark and light gray; VP1–4; blue, green, red, yellow; palmitic acid as a pocket factor, orange).

protect 50% of the cell cultures against challenge ranged from >72408 to 1600 for Sabin and WT serotypes 1, 2, and 3[33]. Here, the neutralizing ability of Fab 9H2 was assessed by $TCID_{50}$, where Fab dilutions 0.02 and 0.002 mg/ml neutralized the virus and prevented infection (see Methods). Affinity measurements were done using biolayer interferometry (BLI) (Supplementary Table 4) with biotinylated virus (Sabin serotype 1 and 3, and Sabin inactivated serotype 2) immobilized on the probe. 9H2 Fab affinity measurements ranged from $9.36 \times 10^{-4}$ to $2.10 \times 10^{-6}$ (M) (Supplementary Table 4).

Competition ELISA binding assays using soluble PVR (sPVR) were performed to assess the ability of the 9H2 monoclonal antibody (mAb) to compete with the receptor. Wild type 1 Mahoney, was captured using purified rabbit antibody that did not interact with either 9H2 or PVR. The ability of 9H2 mAb or sPVR to bind WTPV1 was confirmed individually by ELISA before assessing competition. To address competition, 0.1 mg/mL of either 9H2 mAb or sPVR was applied to the anchored virus before applying the opposite ligand to the wells. 9H2 mAb was able to outcompete sPVR, which was already bound to the virus. Consistent with this finding, sPVR was unable to outcompete and replace 9H2 mAb that bound virus first (Supplementary Fig. 7).

## Discussion

There are no significant structural differences between the live and inactive virus particles of each serotype. Even when comparing wild type to Sabin for types 1 and 2, the differences are negligible (9 and 12 differences). The footprint differences are smaller in number compared to the differences between serotypes and occur adjacent to universal contacts. For each inactive virus footprint, the most similar contacts are found in the corresponding live virus (Supplementary Fig. 6 and Supplementary Table 2). This finding suggests that the inactive virus particle would have similar immunogenicity and could generate cross-neutralizing antibodies as a live vaccine.

The number of contacts does not correlate to the 9H2 neutralization, since types 1 and 2 are neutralized equally well, and type 3 is neutralized slightly less effectively. There is no significant difference among types in the buried surface area of bound 9H2. However, Type 1

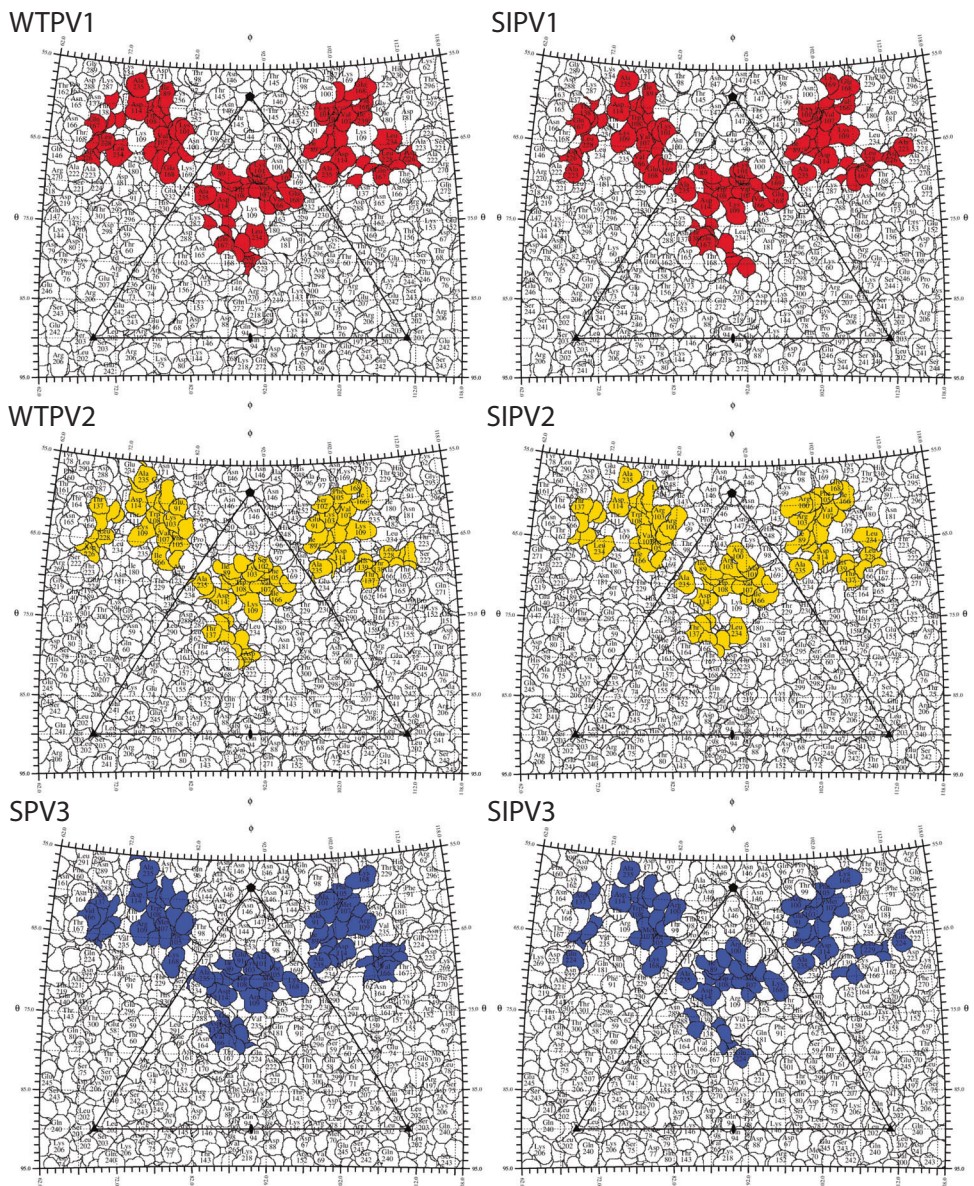

**Fig. 5 | 9H2 footprint roadmaps.** Road maps of the virus surface are represented as a quilt of amino acids, shown as a projection with the icosahedral asymmetric unit indicated by the triangular boundary[58]. Residues for live (left) and inactive virus (right) comprising the footprint of Fab 9H2 are colored according to the serotype of poliovirus in complex: red, serotype 1; gold, serotype 2; blue, serotype 3.

and 2 have three residues in common that type 3 lacks: VP1 residues 166 and D226 that map to the north canyon rim; and VP1 P282 on the south canyon rim (Supplementary Figs. 6 and 8). Of these contacts common to types 1 and 2, two are sequence conserved among all three strains: VP1 D226 and P282. This comparison suggests among the three differences, there are key residues involved in 9H2 neutralization effectiveness, that may be conferred by binding affinity.

9H2's footprint contains a binding epitope the targeting of which allows the neutralization of all three poliovirus serotypes[33]. Antigenic sites N-Ag I-III have been established over several decades by analyzing escape mutants to murine monoclonal antibodies generated by vaccination in rodent systems[23]. The universal contacts of the 9H2 footprints of the 6 complexes (live and inactivated virus particles of the three poliovirus serotypes) have no overlap with any of the N-Ag sites, with individual complex footprints overlapping with up to a few residues in N-Ag I (VP1) or N-Ag II (VP2) (Fig. 6).

While the 9H2 epitope is significantly different from those described in the rodent models, it is similar to the epitopes of both the A12 monoclonal chimpanzee-human chimera[24, 33] and that identified with dromedary nanobodies PVSP6A, PVSS8A, PVSP19B, PVSS21E, and PVSP29F[25]. Binding at such a similar epitope does not guarantee similar cross-neutralizing potential. Both 9H2 and A12 bind and neutralize serotype 1 and 2; however, A12 does not neutralize type 3 and 9H2 does, albeit less effectively. A12 binds serotypes 1 and 2 with different modes of binding, while 9H2 has the same binding mode on all three serotypes[24]. Members of the dromedary nanobody panel can only neutralize serotype 1. Although we cannot identify the exact epitope of the A12 antibody due to the resolution limits at the time (12 Å)[24], it is known that 9H2 competes with A12 for binding to the capsid[33]. The nanobody panel's common contacts have significant overlap with 9H2 contacts for both VP1 and 2, but lack contacts to VP1 88–89 and VP3 235, all universal contacts for 9H2.

The 9H2 footprint overlaps that of PVR but seems to trigger pocket factor release only from serotype 1, indicating that 9H2 lacks contacts essential to PVR function in serotypes 2 and 3 (Table 3). Despite serotype 1 lacking pocket factor, none of the

**Table 1 | Capsid contact comparison**

|      | WTPV1 | Common | WTPV2 |
|------|-------|--------|-------|
|      | WTPV1 | Common | WTPV2 |
| VP1 | 90 101 104 234 239 280 | 87 88 89 102 105 106 107 108 114 166 168 226 227 228 282 | 91 103 109 |
| VP2 | 167 | 139 140 142 | 138 172 |
| VP3 |  | 233 235 |  |
|      | WTPV1 | Common | SPV3 |
| VP1 | 90 104 166 226 234 280 282 | 87 88 89 101 102 105 106 107 108 114 168 227 228 239 | 91 103 109 115 |
| VP2 |  | 139 140 142 167 | 138 141 |
| VP3 |  | 233 235 |  |
|      | WTPV2 | Common | SPV3 |
| VP1 | 166 226 282 | 87 88 89 91 102 103 105 106 107 108 109 114 168 227 228 | 101 115 239 |
| VP2 | 172 | 138 139 140 142 | 167 |
| VP3 |  | 233 235 |  |

Differences in the 9H2 footprint residues are distinguished in these live Fab-viral complexes. Residues are numbered from alignment to the WTPV1 sequence.

**Table 2 | 9H2 contact comparison**

|      | WTPV1 | Common | WTPV2 |
|------|-------|--------|-------|
|      | WTPV1 | Common | WTPV2 |
| Heavy chain | 81 119 120 | 121 122 123 124 125 127 128 129 130 | 74 131 |
| Light chain | 73 | 45 46 50 74 114 115 |  |
|      | WTPV1 | Common | SPV3 |
| Heavy chain | 119 | 81 120 121 122 123 124 125 127 128 129 130 | 74 131 |
| Light chain |  | 45 46 50 73 74 114 115 | 44 52 |
|      | WTPV2 | Common | SPV3 |
| Heavy chain |  | 74 121 122 123 124 125 127 128 129 130 131 | 81 120 |
| Light chain |  | 45 46 50 74 114 115 | 44 52 73 |

Differences in the 9H2 paratope residues are listed comparing the paratope between serotypes.

viruses in complex showed any conformational change in the GH loop which has been reported in another enterovirus incubated with a receptor that lost PF[42]. In addition for all three viruses, the region around VP1 237 remains in the doorstop UP conformation[15] as reported previously as a conformational change observed upon PVR binding. When considering the live virus-Fab complexes, VP1 L234 is found only in the serotype 1 footprint and is critical to PF release (Table 3). Contact residues unique to serotype 1 and not conserved in serotypes 2 or 3 might also play a role (VP1 A106, V107, and E168; VP2 H142) (numbered according to WTPV1 sequence) (Table 3). Conversely, the overlap also identified residues within the PVR footprint that are not in the 9H2 footprint, which are likely essential to trigger critical entry steps. So, how is 9H2 able to neutralize any serotype of poliovirus? In comparing the binding footprint of 9H2 to that of poliovirus receptor CD155 from Strauss et al. 2015 a clear hypothesis emerges. The left half of the PVR footprint directly overlaps with the right half of the 9H2 footprint (Fig. 6), encompassing at least 3 Universal Contact residues and several other non-Universal Contacts on a 9H2-virus complex-dependent basis. This overlap indicates that 9H2 can bind and block PVR attachment to the virus, which is in agreement with the findings from 9H2 mAb and sPVR competition assays where 9H2 was found to outcompete sPVR.

9H2 Fabs are angled away from each other, and their C termini of the variable binding domain have a distance of 74 Å from each other. This geometry rules out the possibility of mAb bivalent interaction on a single capsid but leaves open the possibility of crosslinking multiple capsids as an additional or alternative way to neutralize the virus[43, 44].

The 9H2 human mAb may represent an opportunity for the development of a biologic. Despite searching, no escape mutant to 9H2 has so far been identified in virus-antibody co-culture[33]. Since 9H2 binding overlaps the poliovirus receptor-binding site with highly conserved residues spread across different capsid proteins (VP1, 2, and 3), it is unlikely that a complete 9H2 antibody-resistant escape mutant will arise. We cannot rule out the remote possibility that the 9H2 may select partial escape mutants, though given the common footprint, escape mutants likely would have lower affinity for PVR[45,46] and thus may transmit less efficiently.

## Methods

### Virus propagation and purification
For each strain of virus, confluent layers of HeLa or Vero cells (ATCC) were infected with MOI of 0.1 to 4 and propagated until ~90% CPE was observed, which took between 10 and 24 h. The virus was then titered by plaque assay in 6-well plates in which 250 μL of serially diluted virus in serum-free DMEM (Corning) was allowed to bind to confluent cells at 37 °C for 45 min and then removed and the cells rinsed with phosphate-buffered saline (PBS). 1.25% agarose in DMEM supplemented with 5% BSA and 1% NEAA was then overlayed on the cells. The 6-well plates were returned to 37 °C for approximately 48 h before fixing the cells with 1:9 37% formaldehyde: ddH2O. The remaining cells were stained with crystal violet and plaques were counted to determine virus titer. These steps were iteratively repeated to expand virus stocks and achieve a stable acceptable titer.

Virus purification was adapted from the previously described protocol[47]. Each virus propagation for purification was conducted in a 10 and 2-stack cell factory (VWR). After infection as described, cell lysate was transferred to bottles, frozen and thawed 3 times, and centrifuged at 10 K RPM for 10 min to pellet cell debris. The debris was resuspended and Dounce homogenized with the addition of 10% NP40 surface-amps detergent (Thermo Fisher). The homogenization was combined with the supernatant and centrifuged at 10 K RPM for 10 min. The resulting supernatant was transferred to a 4 L beaker, PEG 8000 was added to 5%, NaCl stock solution was added to 0.5 M, and stirred overnight at 4 °C. The mixture was centrifuged at 10 K, for 10 min to pellet the PEG aggregated virus, which was resuspended in a minimal amount of 50 mM HEPES pH 8, 200 mM NaCl, and 3 mM MgCl2 buffer. MgCl$_2$ 0.5 M was added to 10% of the sample volume followed by DNAse I (Spectrum) and SDS 10% to 5% of the sample volume and incubated at RT for 30 min. Per mL of sample, 0.8 mg trypsin and 0.15 mL of 0.5 M EDTA, pH 9.5 was added and incubated at 37 °C 10 min. N-Lauryl Sarcosine 10% was added to 10% of the sample volume. pH was then raised with ammonium hydroxide as necessary until the DMEM pH indicator returned to pink/red. This sample was then transferred to 50.2 Ti Beckman tubes and 2 mL 30% sucrose in the sample buffer layered beneath it. Tubes were transferred to the Beckman 50.2 Ti rotor for ultracentrifugation at 48 K RPM for 2 h at 4 °C with the slow brake. The supernatant was discarded and the viral pellet rehydrated in 1 mL sample buffer at 4 °C overnight. The fully resuspended virus was applied to a continuous 10–40% K-tartrate gradient, balanced in the SW41 rotor, and ultracentrifuged at 36 K RPM for 1.5 h at 4 °C with no brake. Resulting virus bands were collected into a syringe by piercing the ultracentrifuge tube with an 18 G needle. Virus was concentrated and the buffer was exchanged by ultracentrifugation and resuspension in 50 mM HEPES pH 8, 200 mM NaCl, and 3 mM MgCl2 buffer. Concentration was estimated by spectrophotometry (DeNovix).

Sabin serotypes 1 and 3 were kindly provided by Konstantin M. Chumakov (Office of Vaccines Research and Review, Division of Viral Products, Laboratory of Method Development, FDA). After data

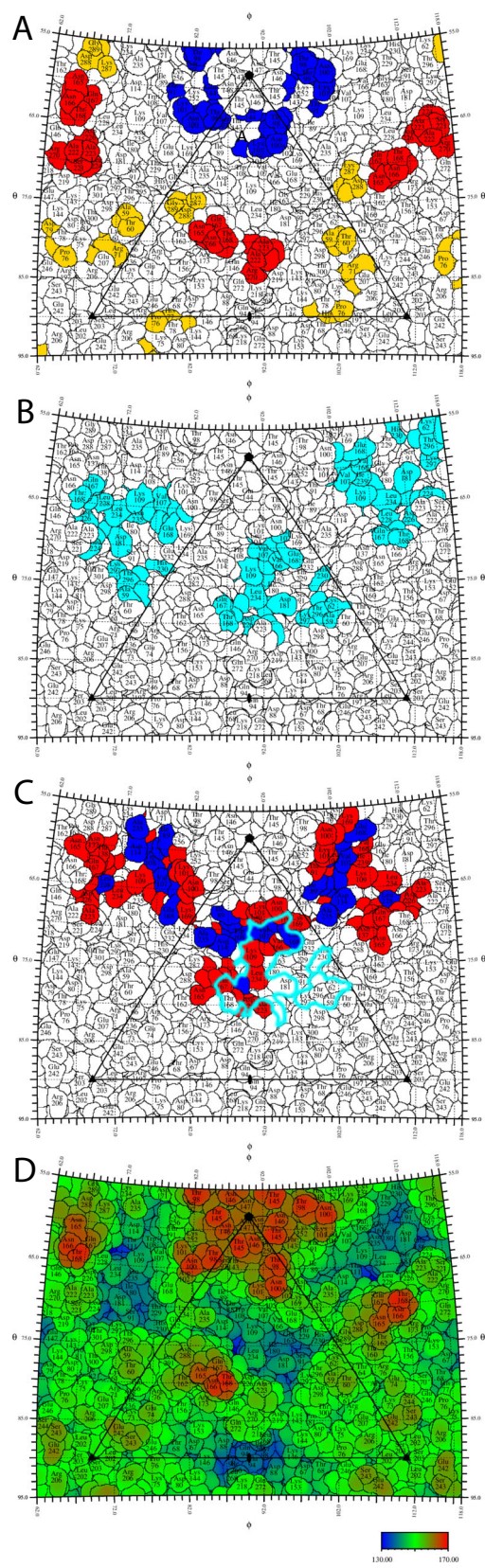

**Fig. 6 | Antigenic sites, PVR footprint, and 9H2 summary footprint.** Amino acids are numbered as sequence-equivalent residues in an alignment to WTPV1.
**A** Antigenic sites (rodent models) as summarized by Fiore et al. 1997[23]; blue, N-Ag I; red, N-Ag II; yellow, N-Ag III. The 9H2 footprint has 5 non-universal contact residues that overlap with N-Ag I & II. **B** The footprint of the PV receptor CD155 (cyan)[15] has a significant overlap with the 9H2 footprint. **C** Summary footprint of 9H2. Blue, universal contacts in all 9H2 complexes (VP1 T88, I89, A106, V107, W108, D114, E168, L228; VP3 A235); red, non-universal contacts. The canyon (black) and CD 155 footprint (cyan) are indicated. **D** Roadmap has been colored by radius from the center of the virus. Color key in Å.

**Table 3 | 9H2 contacts that overlap receptor**

| VP1 | 102 105 **106 107** 109 162 |
|---|---|
| | 166 167 **168** 205 214 224 |
| | 226 228 <u>234</u> 235 237 239 |
| | 295 296 297 |
| VP2 | 139 142 167 168 |
| VP3 | 59 62 176 178 181 182 |
| | 183 184 230 |

PVR contacts with the virus surface. Highlighted residues overlap with the 9H2 footprint whereas non-highlighted residues are not in contact with 9H2[15]. Thus, non-highlighted residues are likely critical to triggering PF release and the next steps of entry. Bolded residues are position-related contacts found in all Fab-virus complexes that do not have conserved amino acid identity. The underlined residue 234, while not a unique contact among all 6 structures, is unique to live virus contacts and may be important for pocket factor release that was observed in serotype 1.

**9H2 mAb and Fab generation.** Mab 9H2 was characterized as reported previously[33] and generously provided to us by Scott Dessain (Lankenau Institute for Medical Research, 100 E. Lancaster Ave., Wynnewood, PA 19096, USA). Neutralization assays were reported previously as >72,408, >72,408, and 36,204 reciprocal dilutions for Sabin poliovirus 1, 2, and 3, respectively[33]. The mAbs were tested in a microneutralization assay, with results reported as the reciprocals of dilutions that protect 50% of the cells against challenge with 100 TCID50 of the indicated strains as described in Puligedda et al.[33].

The 9H2 fragment antigen-binding (Fab) region was digested using the Pierce Fab Micro Preparation Kit (Thermo Fisher) and purified on protein G columns (Thermo Fisher). Sequencing of the heavy and light chains of the mAb was done as reported previously[33]. Fab was concentrated with tabletop centrifuge micro spin columns (Millipore).

**Biolayer interferometry.** The virus was prepared for biolayer interferometry (Fortebio BLItz/Sartorius N1) by biotinylation with the EZ-Link Micro Sulfo-NHS-Biotinylation Kit (Thermo Fisher). Biotinylated virus (SPV1, SIPV2, SPV3) and Fab were each diluted 1:2 into BLItz Kinetics Buffer; 50 mM HEPES pH 8, 200 mM NaCl, and 3 mM MgCl2 buffer with 0.08% Tween 20 and 0.4% BSA. The prepared virus was loaded onto streptavidin probes (Sartorius) using the default advanced kinetics experiment pipeline for the instrument. Kinetics constants were calculated via the analysis tool on the Sartorius software and results are reported in Supplementary Table 4.

**TCID$_{50}$ Fab-PV neutralization assay.** The TCID$_{50}$ neutralizing tests were run in triplicate. Vero cells were grown to >90% confluency in DMEM, 1% NEAA, 10% FBS in 96 well plates. 50 μL 1400 pfu/mL SPV3 virus inoculum was incubated with 50 μL of a range of 9H2 Fab dilutions (0.02 to $0.2 \times 10^{-8}$ mg/mL) in DMEM, 1% NEAA for 30 min RT prior to application to PBS rinsed plates. After an hour of incubation at 32 °C 50 μL DMEM, 1% NEAA, and 5% FBS was added to each well. Plates were monitored continuously for 72 hrs at 32 °C and inspected for CPE daily.

collection, all wild-type PV1 and 2 stocks, inoculum, and associated materials were destroyed in coordination with the CDC and according to their instructions. The inactivated Sabin vaccine virus was supplied by the Beijing Institute of Biological Products, China, was exchanged against 50 mM HEPES pH 8, 200 mM NaCl, and 3 mM MgCl$_2$ buffer, and concentrated as described above.

**Production of soluble poliovirus receptor (sPVR).** Recombinant sPVR was produced using the FreeStyle 293 expression system. Cells were propagated in GIBCO FreeStyle 293 Expression medium. Three hundred micrograms of plasmid p3DPVR/IRES/GFP/MP8DNA, encoding the three external IgG-like domains of PVR conjugated to a histidine epitope[48], was introduced into FreeStyle 293 cells using polyethyleneimine. Cultures were supplemented with 2.2 mM valproic acid the following day. The supernatant was collected 7 days later and clarified by centrifugation (10,000 × *g*; 5 min). Following buffer exchange, enriched supernatant was mixed with loading buffer (final concentration 50 mM NaPO₄, 50 mM NaOH, pH 8.3 3 mM imidazole) and loaded onto a HisTrap FF 1 mL column using an AKTA. The column was washed with 10 volumes of wash buffer (10 mM imidazole in phosphate-buffered saline (1x PBS, 20 mM NaPO₄, 150 mM NaCl, pH7)) at a flow rate of 0.5 mL/min. Bound protein was eluted with elution buffer (50 mM imidazole, 1x PBS) at a flow rate of 0.5 mL/min. Fractions containing sPVR were pooled and dialyzed overnight against buffer B (20 mM HEPES-NaOH, pH 8, 20 mM NaCl). The presence and purity of recombinant sPVR were confirmed by 10% SDS-polyacrylamide gel electrophoresis and by western blot analysis (data not shown). Specific activity was defined as 150 mL of a protein capable of neutralizing $10^3$ plaque-forming units of WTPV1 in a 300 mL reaction volume.

**sPVR virus neutralization by plaque assay.** To confirm function, purified recombinant sPVR was diluted serially 2-fold in Dulbecco's PBS and added to $10^3$ PFU of WTPV1. Neutralization was assessed by plaque assay. HeLa cells were seeded on 35-mm plates and grown to approximately 70% confluence at the time of plaquing. Next, 100-mL portions of serial 10-fold virus dilutions were incubated with cells for 1 h at 37 °C. A single overlay consisting of 2 mL of Dulbecco's modified Eagle's medium (DMEM), 0.8% bacto agar, 0.1% bovine serum albumin, 40 mM MgCl2, and 10% bovine calf serum was added. The cells were incubated at 37 °C for 2 days and developed by using 10% trichloroacetic acid and crystal violet.

**Enzyme-linked immunosorbent assay (ELISA) binding assay.** Enzyme-linked immunosorbent assays (ELISA) were done in flat-bottomed microtiter plates (Nunc Maxisorb, Fisher,44-2404-21). The wells were coated with 100 ng of purified anti-WTPV1 rabbit IgG diluted in 100 mM bicarbonate/carbonate buffer (pH 9.6) and incubated at 4 °C overnight. Unbound capture antibody was removed, and wells were washed using 1x Tris-buffered saline (TBS)-Tween 20 at room temperature and blocked with a solution of 6% fetal bovine serum in 1x TBS-Tween 20 at room temperature for 3 h. Purified human monoclonal antibody 9H2, recombinant sPVR (for controls), or $10^5$ PFU of WTPV1 (experimental setup) (diluted in 1x TBS-Tween 20) were added to the antibody-coated wells, incubated at 4 °C overnight, and unbound antibody, soluble receptor or virus was removed as described above.

For the experimental conditions captured virus was incubated in the presence of 100 mg 9H2 or recombinant sPVR diluted in 1x TBS-Tween 20 at 4 °C overnight. Unbound antibody or soluble receptor was removed, and wells were washed using 1x Tris-buffered saline (TBS)-Tween 20 at room temperature and incubated in blocking solution at room temperature for 3 h.

Subsequently (for both control and experimental conditions) serial 2-fold dilutions of human monoclonal antibody 9H2 or purified recombinant sPVR from an initial concentration of 1 mg/mL each were diluted in 1x TBS-Tween 20 and added to wells in which the virus was previously bound to the other specimen and incubated at 4 °C overnight. The unbound antibody or recombinant soluble receptor was removed, and the wells were washed using 1x TBS-Tween 20. Wells were then incubated in the presence of secondary antibodies diluted 1 to 5,000 in blocking solution for 90 min; either (1) anti-human Fc IgG-conjugated to horseradish peroxidase (Invitrogen) or (2) anti-his-conjugated to horseradish peroxidase (Invitrogen). The wells were washed, and antibody binding was detected by the addition of the TMB substrate (SureBlue, Seracare) and the reaction was stopped by adding of 100 μL TMB STOP-reagent (Sigma). The optical density (OD₄₅₀) was determined within 15 min using a microplate reader (Tecan).

**Sample preparation and Cryo-EM data collection.** Poliovirus-Fab complexes were made by incubating 9H2 Fab and virus together at a ratio of 3:1 Fab per binding site for 30 min at RT. For the vitrification of each sample, 3.5 μL of the complex was applied to a freshly glow-discharged 2/1 Cu Quantifoil grid (Quantifoil Micro Tools GmbH) with a 3–4 nm continuous carbon coating. The sample was incubated on the grid 1 min, blotted, and vitrified on the Mk III Vitrobot (Thermo Fisher) at 4 °C with 100% humidity. If the concentration was low, the sample was applied and blotted multiple times. Cryo-EM datasets were collected at 200 kV with a Talos Arctica or at 300 kV with a Titan Krios microscope (Thermo Fisher) equipped with a spherical aberration corrector at the Huck Institute for Life Sciences cryo EM Facility (Supplementary Table 1). EPU (E Pluribis Unum) software was used for automated single particle data acquisition with a defocus range of −0.5 to −2 (Supplementary Table 1). On the Krios, data were recorded using a Falcon 3 detector with a nominal magnification of x59,000 or x75,000, yielding a final pixel size of 1.1 and 0.88 (Supplementary Table 1). The Talos Arctica equipped with a Falcon 4 direct electron detector was operated at a magnification of x120,000 yielding a final pixel size of 1.2 (Supplementary Table 1).

**Image processing.** All steps of single particle reconstruction for the generation of the icosahedral map and local resolution mapping were conducted in Cryosparc[49]. Icosahedral Extraction and Correlative Classification (ISECC) were performed[34]. ISECC was written for compatibility with RELION 3.1[50], with additional features for correlative subparticle analysis (https://github.com/goetschius/isecc/).The subparticles were extracted, classified, relevant classes pooled, and subparticle density recombined into the icosahedral maps using symbreak within ISECC[34]. Refined maps were postprocessed in DeepEMhancer[35].

**Model building and contact identification.** A model for the 9H2 Fab was generated using SabPred AbodyBuilder[38]. Models for the capsid (PDB ID 1HXS, 1EAH, and 1PVC)[5,36,37] (Supplementary Table 1) were used to initiate the build. Each model was modified as necessary in Coot[51] to add missing residues or change residues to the Sabin strain sequences. The build was conducted first in ISOLDE[52] and subsequently in PHENIX[53] before validation with MolProbity[54] (Supplementary Table 1). Palmitic acid from 1HXS was fitted into pocket factor density. While the pocket factor density was too ambiguous alone to identify the exact pocket factor identity, palmitic acid was selected as it is the most common pocket factor in enteroviuses[55]. Virus-to-Fab contacts were identified as residues having atoms separated by less than 0.4-Å van der Waal's radius[56]. For comparison and figures, contacts were plotted in Chimera or ChimeraX[56,57], With roadmaps generated in RIVEM[58]. The map examples for quality of fit to density were defined using ChimeraX's Map Zone tool[57]. Sequence alignment and comparison were done in Clustal Omega[59]. Chimera was used to assess 9H2 buried surfaces and RMSD.

## Data availability
The cryo EM maps of the 9H2-poliovirus complexes have been deposited in the EM database (EMD-27943, EMD-27948, EMD-27947, EMD-27951, EMD-27950, and EMD-27949) (WTPV1, WTPV2, SPV3, SIPV1, SIPV2, and SIPV3, respectively) (http://www.emdatabank.org). The

coordinates for the 9H2-poliovirus complex atomic models have been deposited in the protein data bank (PDB-8E8L, PDB-8E8S, PDB-8E8R, PDB-8E8Z, PDB-8E8Y, and PDB-8E8X) (WTPV1, WTPV2, SPV3, SIPV1, SIPV2, and SIPV3, respectively) (https://www.rcsb.org). Source data are provided as a Source Data file. Source data are provided in this paper.

## Code availability

Our custom software Icosahedral Subparticle Extraction and Correlative Classification (ISECC) is available on Github https://github.com/goetschius/isecc.

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

## Acknowledgements
Funding was provided by the Pennsylvania Department of Health Commonwealth Universal Research Enhancement (CURE) funds. Research reported in this publication was supported by the Office of the Director, NIH, under award number S10OD026822-01 (S.L.H.) and NIH 1 R01 AI107121-01 (S.L.H.), as well as PATH support through the Gates Foundation grant OPP1135787 (K.M., R.W.).

## Author contributions
A.J.C., K.M.C., K.M., and S.L.H. conceived the study. A.J.C. conducted the cell culture, virus propagation and purification, biochemistry, and virology. A.R. performed the ELISA binding competition assay and interpretation. R.W., K.M.C., S.D., R.D.P., and K.M. provided reagents. J.E.F., V.T., and N.M.D. assisted with some biochemistry steps. C.M.B. and S.H.C. prepared the sample for cryo-EM data collection and collected the cryo-EM data. D.J.G. designed and developed the custom software. A.J.C. solved the structures, refined the maps, built the models, calculated, and interpreted the results. J.E.F., D.J.G., and H.L. assisted with reconstruction. A.J.C. and S.L.H. interpreted the data and wrote the paper.

## Competing interests
The authors declare no competing interests.
