## [Peer Review File · Nature Communications]

REVIEWER COMMENTS

Reviewer #1 (Remarks to the Author):

High resolution cryo EM structures show a human monoclonal antibody binds into the canyon to neutralize all three serotypes of poliovirus by Charnesky et al. The authors describe high resolution structures of a serotype cross-neutralizing human monoclonal antibody binding to the canyon of poliovirus. There are previous reports of similar antibodies that bind to a similar region albeit at 12A resolution, so this is the highest resolution structure of canyon binding antibody to date. There is an outbreak of poliovirus in New York so this report comes in a timely fashion. The knowledge here however, is incremental, and hence, this reviewer feels it is likely not of interest to the general readers and is more suited to specialized journals.

Major comments

(1) Regarding the interpretation of the density map, the validation reports of all structures suggest that some geometries are not good and almost all residues fit poorly into the cryoEM map. Authors should really check the fit of their structures into the density map. Authors should also show zoom-in fitted models in the density maps. Hence this reviewer cannot be certain the interpretation of the structures is correct, as the validation indicates severe problems with the model.

(2) Supplemental Table 1:

a. What does model resolution mean? Is it model/map correlation/FSC? How can the model resolution be better than the resolution of the cryoEM maps?

b. Model resolution range – how can resolution be better than the cryoEM maps again? The reported numbers are impossible, as they are better than the achievable Nyquist resolution limit.

(3) Authors described that they did symmetry expansion and averaging of these subparticles and they refer it to figure 1. However, these maps and structures are not shown in Figure 1.

(4) As the structures are not particularly novel, the authors should conduct some biological assays to test:

a. For affinity of the antibody to different serotypes and correlate to their binding sites.

b. Competitive binding assay of antibody with CD155 receptor to see if antibodies could block receptor binding.

c. Assay showing inhibition of viral attachment to cells.

Minor comments

(1) Line 166: Total 9H2 contacts on the virus surface range from 15-28 ?residues? or Angstrom?

(2) Figure 4: Please outline where the canyon floor is.

(3) Figure 1C and S1C: FFC should be FSC.

Reviewer #2 (Remarks to the Author):

Poliomyelitis is one of the earliest documented human diseases that was also the target of one of the earliest successful vaccine development efforts, leading to its eradication on much of the planet. However, as demonstrated recent events, the disease is by no mean full eradicated and it can easily emerge as a pathogen in non-vaccinated populations in both underdeveloped and developed countries. As such, there is a continued need for treatment and epidemic control measures to prevent the spread of poliovirus. This has historically been done by the WHO using individual vaccine strains targeting the three serotype strains of the virus, which must be separately produced and then be combined as needed for immunization efforts. This is a cumbersome juggling for the WHO and it would be far preferable to have a single immunization or neutralization strategy that would address all the poliovirus strains.

In this work, Charnesky et al. address this specific issue by solving the structures of a

broadly reactive anti-poliovirus antibody, 9H2 first isolated in 2017, in complex with virus particles from all three poliovirus strains. These structures reveal a number of key observations that would facilitate the use of 9H2 and its potential derivatives as universal anti-poliovirus biologicals. First, the antibody binds to the same region of all the serotypes in essentially the same geometry, i.e. the binding is not affected by differences in the virus capsid protein sequences. Second, the 9H2 binding site has significant molecular overlap with the CD155 receptor binding site, leading to the conclusion that 9H2 functions as a neutralizing antibody by sterically interfering with receptor binding to prevent virus uptake by cells. Third, the "pocket factor" lipid that is an integral part of the virion structure remains bound in the presence of 9H2, meaning the capsid are not destabilized by antibody binding as is often the case. This last observation is likely a key aspect of 9H2 function in that partially receptor-bound particles are not predisposed to disassembly upon cell entry, meaning the infectivity of those virus particles that do get into cells is also reduced.

Altogether, the three core findings for this work mark an important contribution to our molecular understanding of 9H2 neutralizing antibody function and set the stage for the further development of optimized antibodies or other antiviral biologicals that can be effective across all three poliovirus serotypes. From a technical standpoint, the structural biology work in the manuscript is excellent and well described, and the structures are well illustrated except for some minor notes below.

One point where the discussion could be enhanced is the discussion of Fab binding contacts and inferred affinities on the different virus serotypes (lines 206-213). This section is highly descriptive but given relatively fixed capsid structures and a common unbound structure for the 9H2 antibody it would seem reasonable to make a somewhat quantitative comparison of these structures. This could perhaps be done by calculating the loss of solvent accessible surface area in the different complexes and seeing if this also correlates with the reduced neutralization of type 3 poliovirus.

Minor issues for author consideration:

- There is inconsistent use of FAb versus Fab in the text
- Yellow color can be difficult to discern and the authors could perhaps consider something more toward an orange for improved visibility (Figures 1, 3, 4, 5, 6).
- the Pocket Factor lipid is difficult to see in Figure 3 due to size, so perhaps draw with small spheres and/or in a more distinct color than black.
- this reviewer encountered errors when opening the supplied PDF files using both Windows and Mac computers, but the MS Word version worked fine. It is not clear if the issue is with the authors formatting or with Nature Publishing Group's PDF generation.

Reviewer #3 (Remarks to the Author):

Polio eradication proved to be a much more difficult task than initially believed, as shown by the recent identification of circulating poliovirus (PV) in London and New York. This makes the development of novel antiviral therapies a top priority. An important role toward this aim is the identification and characterization of potent neutralizing antibodies (nAbs). The manuscript of Charnesky and colleagues describes the mode of action of 9H2, a potent nAb, by solving a series of high resolution structures describing its interaction with the full capsid for all PV strains in both the active and the inactivated form.

In order to increase the resolution of the contact region, the authors used an algorithm in which all symmetry related positions on each capsid were separately extracted and classified. The reconstructed maps show that 9H2 footprint overlaps with the receptor

binding site for all PV strains. The binding is practically identical for each complex, for both wild type and inactivated PV. Minor differences in the position of VP1 loops are discussed. The Ab makes contacts on both sides of the canyon and also with some residues of the canyon floor.

Overall, the study is very well constructed, the results are definitive and clearly presented. The manuscript will be of interest for structural virologists as well for scientists interested in developing PV therapies.

I think that the manuscript will be improved if the following minor points will be addressed:

- Due to the general audience of this journal, readers might not be familiar with PV terminology. I suggest authors to offer some explanation regarding the D antigenic conformation, the role of the pocket factor in the stability of the capsid, etc.**
- Characterization of the 9H2 antibody was reported in a previous article. However, I suggest that some values regarding its potency to be included in the current manuscript.**
- In the roadmap figures, please indicate the canyon and CD155 footprints. The author can use cyan for the receptor (as in 6c) and black for the canyon.**
- Probably Fig 5 can be used as supplementary material.**

Response to Reviewers

Note: Authors responses are in blue font.

All reviewers' concerns have been carefully addressed below.

Reviewer #1 (Remarks to the Author):

High resolution cryo EM structures show a human monoclonal antibody binds into the canyon to neutralize all three serotypes of poliovirus by Charnesky et al.

The authors describe high resolution structures of a serotype cross-neutralizing human monoclonal antibody binding to the canyon of poliovirus. There are previous reports of similar antibodies that bind to a similar region albeit at 12A resolution, so this is the highest resolution structure of canyon binding antibody to date. There is an outbreak of poliovirus in New York so this report comes in a timely fashion. The knowledge here however, is incremental, and hence, this reviewer feels it is likely not of interest to the general readers and is more suited to specialized journals.

Major comments

(1) Regarding the interpretation of the density map, the validation reports of all structures suggest that some geometries are not good and almost all residues fit poorly into the cryoEM map. Authors should really check the fit of their structures into the density map. Authors should also show zoom-in fitted models in the density maps.

All the builds have been reworked and refined, in one model a numbering error in the heavy chain of the Fab has been fixed, and new validation reports have been generated for inclusion in the resubmission. Zoomed views of the built models in the cryo EM density have been added to Figure 1 as a panel E and Supplemental Figure 1, panel E.

Hence this reviewer cannot be certain the interpretation of the structures is correct, as the validation indicates severe problems with the model.

(2) Supplemental Table 1:

a. What does model resolution mean? Is it model/map correlation/FSC? How can the model resolution be better than the resolution of the cryoEM maps?

b. Model resolution range – how can resolution be better than the cryoEM maps again? The reported numbers are impossible, as they are better than the achievable Nyquist resolution limit.

The journal suggests inclusion of values for model resolution d_{model} and model resolution d_{FSC} . Both criteria are now defined in the table. Briefly, model resolution d_{model} is the resolution of the map calculated from the final model that maximizes the correlation to the experimental map. Model resolution d_{FSC} is the resolution cutoff at which the model and experimental map Fourier coefficients are most similar. Both definitions are referenced with Afonine et al 2018. Both values are independent of the map resolution and thus, may exceed that of the map resolution which is limited by Nyquist as commented by the Reviewer.

(3) Authors described that they did symmetry expansion and averaging of these subparticles and they refer it to figure 1. However, these maps and structures are not shown in Figure 1.

Subparticles are now included and illustrated as Supplemental Figure 2.

(4) As the structures are not particularly novel, the authors should conduct some biological assays to test:

- a. For affinity of the antibody to different serotypes and correlate to their binding sites.
- b. Competitive binding assay of antibody with CD155 receptor to see if antibodies could block receptor binding.
- c. Assay showing inhibition of viral attachment to cells.

All three assays are now included in the Results and Discussion sections. The findings add significantly to the manuscript and we are grateful to the Reviewer for this suggestion.

Minor comments

(1) Line 166: Total 9H2 contacts on the virus surface range from 15-28 ?residues? or Angstrom?

This has been clarified.

(2) Figure 4: Please outline where the canyon floor is.

The canyon has been outlined.

(3) Figure 1C and S1C: FFC should be FSC.

Thank you, this typo has been corrected.

Reviewer #2 (Remarks to the Author):

Poliomyelitis is one of the earliest documented human diseases that was also the target of one of the earliest successful vaccine development efforts, leading to its eradication on much of the planet. However, as demonstrated recent events, the disease is by no mean full eradicated and it can easily emerge as a pathogen in non-vaccinated populations in both underdeveloped and developed countries. As such, there is a continued need for treatment and epidemic control measures to prevent the spread of poliovirus. This has historically been done by the WHO using individual vaccine strains targeting the three serotype strains of the virus, which must be separately produced and then be combined as needed for immunization efforts. This is a cumbersome juggling for the WHO and it would be far preferable to have a single immunization or neutralization strategy that would address all the poliovirus strains.

In this work, Charnesky et al. address this specific issue by solving the structures of a broadly reactive anti-poliovirus antibody, 9H2 first isolated in 2017, in complex with virus particles from

all three poliovirus strains. These structures reveal a number of key observations that would facilitate the use of 9H2 and its potential derivatives as universal anti-poliovirus biologicals. First, the antibody binds to the same region of all the serotypes in essentially the same geometry, i.e. the binding is not affected by differences in the virus capsid protein sequences. Second, the 9H2 binding site has significant molecular overlap with the CD155 receptor binding site, leading to the conclusion that 9H2 functions as a neutralizing antibody by sterically interfering with receptor binding to prevent virus uptake by cells. Third, the “pocket factor” lipid that is an integral part of the virion structure remains bound in the presence of 9H2, meaning the capsid are not destabilized by antibody binding as is often the case. This last observation is likely a key aspect of 9H2 function in that partially receptor-bound particles are not predisposed to disassembly upon cell entry, meaning the infectivity of those virus particles that do get into cells is also reduced.

Altogether, the three core findings for this work mark an important contribution to our molecular understanding of 9H2 neutralizing antibody function and set the stage for the further development of optimized antibodies or other antiviral biologicals that can be effective across all three poliovirus serotypes. From a technical standpoint, the structural biology work in the manuscript is excellent and well described, and the structures are well illustrated except for some minor notes below.

One point where the discussion could be enhanced is the discussion of Fab binding contacts and inferred affinities on the different virus serotypes (lines 206-213). This section is highly descriptive but given relatively fixed capsid structures and a common unbound structure for the 9H2 antibody it would seem reasonable to make a somewhat quantitative comparison of these structures. This could perhaps be done by calculating the loss of solvent accessible surface area in the different complexes and seeing if this also correlates with the reduced neutralization of type 3 poliovirus.

We have calculated and added the buried surface area to address the Reviewer's comments Line 252. Comments concerning these buried surface areas are now included in Discussion, line 386-387 where it is noted that they are not significantly different among types.

Minor issues for author consideration:

- There is inconsistent use of FAb versus Fab in the text

Thank you, this was changed so that Fab is used throughout.

- Yellow color can be difficult to discern and the authors could perhaps consider something more toward an orange for improved visibility (Figures 1, 3, 4, 5, 6).

This has been changed to a darker gold color.

- the Pocket Factor lipid is difficult to see in Figure 3 due to size, so perhaps draw with small spheres and/or in a more distinct color than black.

The PF has been rendered thicker and displayed in magenta.

- this reviewer encountered errors when opening the supplied PDF files using both Windows and Mac computers, but the MS Word version worked fine. It is not clear if the issue is with the authors formatting or with Nature Publishing Group's PDF generation.

Reviewer #3 (Remarks to the Author):

Polio eradication proved to be a much more difficult task than initially believed, as shown by the recent identification of circulating poliovirus (PV) in London and New York. This makes the development of novel antiviral therapies a top priority. An important role toward this aim is the identification and characterization of potent neutralizing antibodies (nAbs). The manuscript of Charnesky and colleagues describes the mode of action of 9H2, a potent nAb, by solving a series of high resolution structures describing its interaction with the full capsid for all PV strains in both the active and the inactivated form.

In order to increase the resolution of the contact region, the authors used an algorithm in which all symmetry related positions on each capsid were separately extracted and classified. The reconstructed maps show that 9H2 footprint overlaps with the receptor binding site for all PV strains. The binding is practically identical for each complex, for both wild type and inactivated PV. Minor differences in the position of VP1 loops are discussed. The Ab makes contacts on both sides of the canyon and also with some residues of the canyon floor.

Overall, the study is very well constructed, the results are definitive and clearly presented. The manuscript will be of interest for structural virologists as well for scientists interested in developing PV therapies.

I think that the manuscript will be improved if the following minor points will be addressed:

- Due to the general audience of this journal, readers might not be familiar with PV terminology. I suggest authors to offer some explanation regarding the D antigenic conformation, the role of the pocket factor in the stability of the capsid, etc.

We have changed some terminology (and removed "D antigen") for clarity and added background and explanation to the introduction.

- Characterization of the 9H2 antibody was reported in a previous article. However, I suggest that some values regarding its potency to be included in the current manuscript.

The effectiveness of 9H2 was added. We have also included new assays on affinity, competition with receptor, and virus infectivity loss.

- In the roadmap figures, please indicate the canyon and CD155 footprints. The author can use cyan for the receptor (as in 6c) and black for the canyon.

The canyon is indicated, and the CD155 binding site outlined in added panel.

- *Probably Fig 5 can be used as supplementary material.*

Figure 5 has been moved to supplemental material.

REVIEWER COMMENTS

Reviewer #1 (Remarks to the Author):

This reviewer #1 has one more comment:

The dmodel and dFSC calculate the resolutions in which the model correlate to the experimental map in real and Fourier spaces, respectively. It's a comparison to the experimental map, therefore, it is not independent of the experiment map. Thus, resolution determined CANNOT be better than the experiment map, and impossible to be better than the achievable Nyquist. The current numbers could indicate that the models are overfitted into the maps, molecules are put in where noises are. From the validation report, the protein side chain outliers are high, and the Q-scores are low, these indicate that side changes have bad geometry or/and the densities around the side chains are poor (resolution is not good enough). This reviewer suggests the authors to go through the fit again, to consider which side chains fit well and which to remove before calculation of dmodel and dFSC.

Reviewer #2 (Remarks to the Author):

In this revision, the authors have nicely addressed the concerns from the first review and there are no major issues with quality of the current manuscript. The work provides significant insights into broadly neutralizing antibody interactions for poliovirus control and will have high impact in the field of understanding virus-antibody interactions.

One minor stylistic issue is that in many places there are numerical values quote to a seemingly inappropriate number of significant digits that do not reflect the typical precision with which the values are known from the data or at which they matter biologically. For example, RMSD values in line 135 are probably realistic to tenths, but not thousandths, of angstroms, surface areas on line 164 are more appropriate to 10s or even 100s of angstroms (1200-1400), and distance as in line 271 are probably better at a one angstrom level (75Å).

Reviewer #3 (Remarks to the Author):

I consider that in the present version the authors have responded all the reviewers' comments.

Second Response to Reviewers

Note: Authors responses are in blue font.

All reviewers' concerns have been carefully addressed below.

REVIEWER COMMENTS

Reviewer #1 (Remarks to the Author):

This reviewer #1 has one more comment:

The d_{model} and d_{FSC} calculate the resolutions in which the model correlate to the experimental map in real and Fourier spaces, respectively. It's a comparison to the experimental map, therefore, it is not independent of the experiment map. Thus, resolution determined CANNOT be better than the experiment map, and impossible to be better than the achievable Nyquist. The current numbers could indicate that the models are overfitted into the maps, molecules are put in where noises are.

Per Table 2 in the Afonine et al 2018 paper (DOI: [10.1107/S2059798318009324](https://doi.org/10.1107/S2059798318009324)) when the resolution of the d_{model} exceeds that of the experimental map (d_{FSC}) the experimental input map has not yet been sharpened. Thus, following the suggestions by the software developers themselves, we used our sharpened maps, and the problem was corrected. The new values are now reported.

From the validation report, the protein side chain outliers are high, and the Q-scores are low, these indicate that side chains have bad geometry or/and the densities around the side chains are poor (resolution is not good enough). This reviewer suggests the authors to go through the fit again, to consider which side chains fit well and which to remove before calculation of d_{model} and d_{FSC} .

As suggested, we revisited the model builds to improve the geometry of the side chains and decrease the sidechain outliers.

To illustrate the quality of our maps, we have added local resolution maps as Figure 2 and SFigure 2.

In accordance with map resolution, some sidechains do extend from density especially in flexible regions, such as the Fab hinge region in the complex maps. However, this does not detract from the overall structural quality, or the biological information obtained. The builds are appropriate to maps of the resolutions reported ranging from 2.5-3.2Å.

It is suitable for the model to be validated without manipulation of sidechains that are not completely contained within the map density. To do otherwise would be to misrepresent the current resolution and quality, which is what validation is supposed to represent.

The Q-scores are appropriate for each map, and to aid in understanding we have compiled a table for our structures. Below please find the resolutions, sidechain outliers (from PDB validation report), and the expected Q-score (output by the Q-score chimera plugin MapQ) and the actual Q-scores measured by MapQ.

	PV1	PV2	PV3	S1	S2	S3
Resolution	3.1Å	2.7Å	2.7Å	3.2Å	2.5Å	2.9Å
Sidechain Outliers	0.4%	1.9%	1.2%	1.0%	0.4%	2.0%
Expected Q-score	0.55	0.64	0.65	0.55	0.69	0.60
Actual Q-score(VP1)	0.61	0.73	0.73	0.62	0.69	0.67

Reviewer #2 (Remarks to the Author):

In this revision, the authors have nicely addressed the concerns from the first review and there are no major issues with quality of the current manuscript. The work provides significant insights into broadly neutralizing antibody interactions for poliovirus control and will have high impact in the field of understanding virus-antibody interactions.

One minor stylistic issue is that in many places there are numerical values quote to a seemingly inappropriate number of significant digits that do not reflect the typical precision with which the values are known from the data or at which they matter biologically. For example, RMSD values in line 135 are probably realistic to tenths, but not thousandths, of angstroms, surface areas on line 164 are more appropriate to 10s or even 100s of angstroms (1200-1400), and distance as in line 271 are probably better at a one angstrom level (75Å).

We thank Reviewer #2 and have corrected all values to significant digits.

Reviewer #3 (Remarks to the Author):

I consider that in the present version the authors have responded all the reviewers' comments.

We appreciate the comments and thank Reviewer #3.

REVIEWERS' COMMENTS

Reviewer #1 (Remarks to the Author):

The molecules fitting has been fixed and this reviewer feels the paper is ready for acceptance.